# STEP : Out-of-Distribution Detection in the Presence of Limited In-distribution Labeled Data

**Zhi Zhou**[1]*, **Lan-Zhe Guo**[1]*, **Zhanzhan Cheng**[2], **Yu-Feng Li**[1][†], **Shiliang Pu**[2]

[1]National Key Laboratory for Novel Software Technology, Nanjing University, Nanjing, China
{zhouz, guolz, liyf}@lamda.nju.edu.cn
[2]Hikvision Research Institute, Hangzhou, China
{chengzhanzhan, pushiliang.hri}@hikvision.com

## Abstract

Existing semi-supervised learning (SSL) studies typically assume that unlabeled and test data are drawn from the same distribution as labeled data. However, in many real-world applications, it is desirable to have SSL algorithms that not only classify the samples drawn from the same distribution of labeled data but also detect out-of-distribution (OOD) samples drawn from an unknown distribution. In this paper, we study a setting called semi-supervised OOD detection. Two main challenges compared with previous OOD detection settings are i) the lack of labeled data and in-distribution data; ii) OOD samples could be unseen during training. Efforts on this direction remain limited. In this paper, we present an approach STEP significantly improving OOD detection performance by introducing a new technique: Structure-Keep Unzipping. It learns a new representation space in which OOD samples could be separated well. An efficient optimization algorithm is derived to solve the objective. Comprehensive experiments across various OOD detection benchmarks clearly show that our STEP approach outperforms other methods by a large margin and achieves remarkable detection performance on several benchmarks.

## 1 Introduction

Deep learning has achieved success in many application scenarios, such as computer vision, speech recognition, natural language processing [10]. The excellent performance typically rely on sufficient supervised information. However, collecting large amounts of well-labeled training data is not always available in real-world applications due to the expensive cost of the labeling process. Therefore, tremendous efforts have been devoted to semi-supervised learning (SSL) [36, 30] which aims to enhance the model performance by exploiting much cheaper unlabeled data, and have shown promising performance [52, 37, 29].

Previous SSL studies [39, 41] typically work on the assumption that unlabeled data and test data are drawn from the same distribution as labeled data. However, it is often the case that such an assumption fails in practical applications [13, 14]. For example, in document classification [9], irrelevant documents readily occur in the testing data leading to high-confidence misclassification. Similar cases commonly appear in other applications, such as medical diagnosis [4] and autonomous driving [8]. In such applications, it is desirable to have SSL algorithms which could not only classify samples from known distributions accurately but also be equipped with the ability to detect out-of-distribution (OOD) samples from unknown distributions precisely.

---

*Contribute to this work equally

[†]Corresponding author

35th Conference on Neural Information Processing Systems (NeurIPS 2021).

OOD detection has been studied for a long history with numerous methods proposed, such as ODIN [31], Mahalanobis [27], DeConf [20], ELOC [43]. These methods perform OOD detection based on the logits of the model or the Mahalanobis distance in the feature space. However, it is hard to adapt these methods to semi-supervised settings because they all rely on massive labeled data. There are some methods associating with unlabeled data, such as UOOD [49], CSI [40], SSD [38] have been proposed recently. These methods assume that the model can obtain sufficient in-distribution (ID) labeled data or ID unlabeled data during the training process. Such an assumption also limits their ability to practical problems.

Therefore, we study a novel setting called semi-supervised OOD detection. Specifically, only a tiny subset of ID labeled data is observed. The other ones remain unlabeled and may belong to ID or OOD. Here, we assume that abundant ID data is contained in the unlabeled data for extracting ID information. This setting is ubiquitous in real-world applications. For example, in web page classification [47], acquiring large numbers of web pages annotated with relevant categories is very expensive, and unlabeled web pages crawled from the Internet according to keywords usually contain irrelevant pages that belong to unseen categories. In medical diagnosis [4], warning users of the model's uncertainty is crucial because any unfaithful diagnosis will bring unimaginable disasters to the patients' health. In ride-sharing liability judgment [15], detecting abnormal orders is of significant value, while collecting training data will meet similar problems stated above. Similar cases often occur in other real-world applications, such as crowdsourcing [45, 28]. There are two main challenges for us compared with previous OOD detection settings. First, both the labeled data and directly available ID data are limited, while sufficient unlabeled data is mixed with ID and OOD samples. Second, OOD samples could be unseen during the training, requiring more stringent generalization of the model.

Focusing on semi-supervised OOD detection, we find that the widely-used Mahalanobis distance is no longer suitable as the confidence score for OOD detection. This is because the necessary covariance matrix $\hat{\Sigma}$ for calculating Mahalanobis distance is hard to estimate accurately with limited ID samples which will severely affect the performance of OOD detection. To alleviate this issue, we propose a novel STEP (STructure-keEP) approach. The idea is to detect OOD samples in a detection-specific space where we maintain the same local topological structures as the original feature space, because the relationships between samples need to be confirmed through local topological structures. We introduce a new objective and optimize it efficiently. The experiments prove that our STEP approach outperforms previous methods by a large margin on diverse data sets.

The contributions of our paper are summarized as follows:

- We propose a practical setting for OOD detection, called semi-supervised OOD detection.

- To alleviate the problem of Mahalanobis distance that the necessary covariance matrix $\hat{\Sigma}$ is hard to be estimated with limited ID samples, we present a new distance calculated in a detection-specific space as OOD confidence scores.

- We evaluate our approach with comprehensive experiments across various OOD detection benchmarks. Our proposal outperforms previous methods by a large margin and achieves remarkable detection performance on several benchmarks.

## 2 Method

### 2.1 Notations and Setting

In the semi-supervised OOD detection setting, we assume that a limited label data set $\mathcal{D}_l = \{(\mathbf{x}_i, y_i)\}_{i=1}^n$ consisting $n$ samples with labels drawn from ID, and an unlabeled data set $\mathcal{D}_u = \{(\mathbf{x}_i)\}_{i=1}^m$ consisting $m$ unlabeled samples drawn both ID and OOD, are accessible during the training phase. We denote the set of ground-truth classes in the labeled data set $\mathcal{D}_l$ and unlabeled data set $\mathcal{D}_u$ as $\mathcal{C}_l$ and $\mathcal{C}_u$, respectively. The labeled samples can be classified into one of $K$ classes denoted by $\mathcal{C}_l = \{c_1, c_2, \ldots, c_K\}$, and the unlabeled samples can be classified into the seen $K$ classes $\mathcal{C}_l$ and some unseen classes denoted by $\mathcal{C}_n = \mathcal{C}_u \backslash \mathcal{C}_l$. The goal is to distinguish whether a sample in $\mathcal{D}_u$ or an unknown testing sample is drawn from ID or not.

## 2.2 Inaccurate Mahalanobis Distance and Our Approach

Mahalanobis distance which is widely used in previous studies [27, 38], has been proven to be a powerful metric in OOD detection. $\mathcal{MD}(\mathbf{x}_i, \mathbf{x}_j)$ denotes the function measuring the Mahalanobis distance between sample $\mathbf{x}_i$ and sample $\mathbf{x}_j$ based on estimated covariance matrix $\hat{\boldsymbol{\Sigma}}$:

$$\mathcal{MD}(\mathbf{x}_i, \mathbf{x}_j) = \sqrt{(\mathbf{x}_i - \mathbf{x}_j)^\top \hat{\boldsymbol{\Sigma}}^{-1}(\mathbf{x}_i - \mathbf{x}_j)} \tag{1}$$

Previous methods mentioned above calculate the minimum Mahalanobis distance between target sample $\mathbf{x}$ and each class center as the confidence score:

$$\text{SCORE}_{\mathcal{MD}}(\mathbf{x}) = \min_{c \in c_1, c_2, ..., c_K} \mathcal{MD}(\mathbf{x}, \mu_c) \tag{2}$$

where $\mu_c$ denotes the center of samples which belong to class $c$ and $\hat{\boldsymbol{\Sigma}}$ is the covariance matrix estimated on all ID samples.

However, $\hat{\boldsymbol{\Sigma}}$ is hard to be accurately estimated in a semi-supervised OOD detection setting because the available ID labeled data set $\mathcal{D}_l$ is insufficient. Inaccurate estimation of $\hat{\boldsymbol{\Sigma}}$ will affect the calculation of Mahalanobis distance. This makes it difficult for the algorithm to distinguish OOD samples and ID samples near the cluster boundary.

Instead of using inaccurate Mahalanobis distance, we decide to learn a $\mathbf{P}$ to project samples into space where a large margin separates ID samples and OOD samples. Inspired by the topological technology [44] used in noisy label problems and cluster assumption [36] used in SSL, we hope that the projected samples can maintain the same local topological structure as the original space while increasing the distance between samples not directly topologically connected. Because of the inaccurate estimation of $\hat{\boldsymbol{\Sigma}}$, we consider that relationship between samples that are not topologically adjacent is uncertain. Their relationships need to be confirmed through each local topological structure. We formulate our goal into the objective:

$$\begin{aligned}
\max_{\mathbf{P}} \quad & \sum_{\mathbf{x}_i, \mathbf{x}_j \in \mathcal{D}_l \cup \mathcal{D}_u} \|\mathbf{P}\mathbf{x}_i - \mathbf{P}\mathbf{x}_j\|_2 \\
\text{s.t.} \quad & \|\mathbf{P}\mathbf{x}_i - \mathbf{P}\mathbf{x}_n\|_2 = \mathcal{MD}(\mathbf{x}_i, \mathbf{x}_n), \\
\text{if} \quad & \mathbf{x}_n \in \mathcal{B}_k(\mathbf{x}_i)
\end{aligned} \tag{3}$$

where, $\mathcal{M}(\mathbf{x}_i, \mathbf{x}_j)$ is the Mahalanobis distance between $\mathbf{x}_i$ and $\mathbf{x}_j$ in the feature space and $\mathcal{B}_k(\mathbf{x}_i)$ is the set of k nearest neighbours of $\mathbf{x}_i$.

Finally, our detection-specific metric can directly calculate as L2 distance in the projected space:

$$\mathcal{N}(\mathbf{x}_i, \mathbf{x}_j) = \|\mathbf{P}\mathbf{x}_i - \mathbf{P}\mathbf{x}_j\|_2 \tag{4}$$

## 2.3 Backbone Pretraining

Our semi-supervised OOD detection task considers OOD detection as a clustering problem based on the feature space. Therefore, reliable feature representations are essential. Benefiting from recent progress on self-supervised learning, we adopt a simple contrastive learning method SimCLR [5] to pre-train our backbone network on the whole dataset $\mathcal{D}_u \cup \mathcal{D}_l$ in an unsupervised fashion. We find that representations obtained by SimCLR have a reasonable ability to distinguish ID and OOD samples. Notably, the learned representations could be not only used for our STEP approach but also used as the initialization of downstream tasks.

## 2.4 Structure-Keep Unzipping

Based on the representations obtained by SimCLR, we further train a $\mathbf{P}$ to project samples into a detection-specific space via our objective formulated in Eq.(3). However, there are two main difficulties: i) Building a KNN graph for extracting topological structure needs $\mathcal{O}(n^2 d^2)$ time complexity to calculate Mahalanobis distance between each pair of samples. This step is very time-consuming because we use an ensemble of representations from each backbone network's layer, and feature dimension $d$ is relatively large. ii) The constraint in Eq.(3) can not be directly optimized.

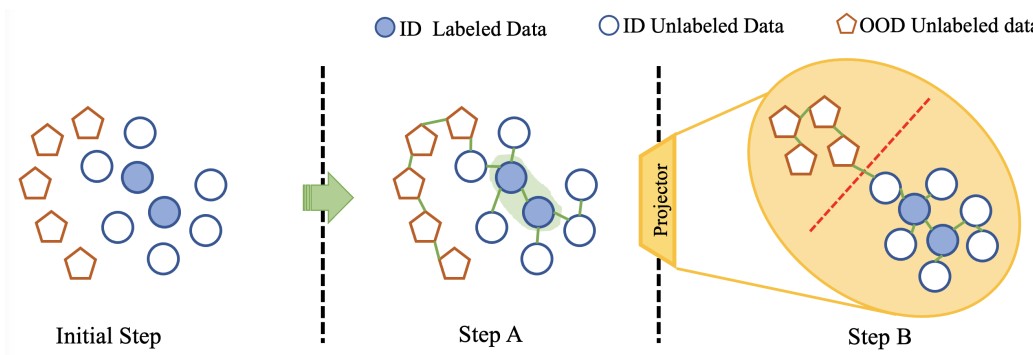

Figure 1: The overall of STEP approach: (a) In initial step, we use contrastive learning to train initial representations. (b) In step A, we estimated statistics information via limited labeled data and extracted topological structure via the KNN algorithm. (c) In step B, we train a $\mathbf{P}$ to project all the samples into a detection-specific space where we can use L2 distances as OOD scores.

First, we transform the process of calculating pairwise Mahalanobis distance into calculating pairwise Euclidean distance in projection space. The time complexity of this step reduces from $\mathcal{O}(n^2 d^2)$ to $\mathcal{O}(n^2 d)$. Specifically, as shown in Eq.(5), we can perform cholesky decomposition on $\hat{\mathbf{\Sigma}}^{-1}$ to get linear projector the $\mathbf{P}_{\mathcal{MD}}$. Then, we multiply all samples by $\mathbf{P}_{\mathcal{MD}}$ to project them into a new space where Euclidean distance equals to original Mahalanobis distance between each pair of samples. There are $n^2$ pairwise Euclidean distances to calculate, and each calculation costs $\mathcal{O}(d)$ time complexity. Therefore, the total time complexity of this step is $\mathcal{O}(n^2 d)$.

$$\mathcal{MD}(\mathbf{x}_i, \mathbf{x}_j) = \sqrt{(\mathbf{x}_i - \mathbf{x}_j)^\top \hat{\mathbf{\Sigma}}^{-1}(\mathbf{x}_i - \mathbf{x}_j)} = \|\mathbf{P}_{\mathcal{MD}}\mathbf{x}_i - \mathbf{P}_{\mathcal{MD}}\mathbf{x}_j\|_2$$
$$\text{s.t.} \quad \mathbf{P}_{\mathcal{MD}}^\top \mathbf{P}_{\mathcal{MD}} = \hat{\mathbf{\Sigma}}^{-1} \tag{5}$$

After converting Mahalanobis distance to Euclidean distance, we can further use the advanced KNN toolkit, such as Faiss [22], to speed up the entire process.

Second, we define $L_{Keep}$ and $L_{Unzip}$ that can be directly optimized to approximately achieve our objective shown in Eq.(3). Both $L_{Keep}$ and $L_{Unzip}$ are shown in Eq.(6):

$$\begin{cases} L_{Keep} & = \max(0, \|\mathbf{P}\mathbf{x}_i - \mathbf{P}\mathbf{x}_n\|_2 - \mathcal{MD}(\mathbf{x}_i, \mathbf{x}_n)), \\ L_{Unzip} & = -\|\mathbf{P}\mathbf{x}_i - \mathbf{P}\mathbf{x}_j\|_2. \end{cases} \tag{6}$$

where $\mathbf{x}_i, \mathbf{x}_j$ are randomly sampled from $\mathcal{D}_l \cup \mathcal{D}_u$, and $x_n$ is randomly sampled from $\mathcal{B}_k(\mathbf{x}_i)$. The final loss to optimize $\mathbf{P}$ is $Loss = L_{Keep} + L_{Unzip}$. The overall of our STEP approach is summarized in Fig.(1), and the pseudo-code of our approach is shown in Algo.(1).

In the detection stage, we directly use the minimum L2 distance between the target sample and each class center in the detection-specific space as the confidence score:

$$Score(\mathbf{x}) = \min_{c \in \{c_1, c_2, ..., c_K\}} \mathcal{N}(\mathbf{x}, \boldsymbol{\mu}_c) \tag{7}$$

where the $\boldsymbol{\mu}_c$ is the center of class $c$ in the original feature space.

## 3 Experiments

### 3.1 Experimental Setup

**In-distribution Data Set.** We use CIFAR-10 and CIFAR-100 [25] as ID data sets in our experiments. They both contain 50,000 training images and 10,000 testing images. The image size of these two data sets is $32 \times 32$. For CIFAR-10, each image belongs to one of 10 classes, and we randomly sample 250 training images as ID labeled data $\mathcal{D}_l$. For CIFAR-100, the size of image classes is 100, and we randomly sample 400 training images as ID labeled data. $\mathcal{D}_l$. We add the remaining training images to the unlabeled data $\mathcal{D}_u$.

---

**Algorithm 1** Training Phase of STEP

---

**Input:** $\mathcal{D}_l$: ID labeled data set; $\mathcal{D}_u$: unlabeled mixed data set; $K$: number of neighbours
**Output:** pre-trained backbone $f_\theta(\cdot)$; projctor $\mathbf{P}$
 1: train backbone $f_\theta(\cdot)$ via contrastive learning on $\mathcal{D}_l \cup \mathcal{D}_u$
 2: estimate $\hat{\boldsymbol{\Sigma}}$ on $\mathcal{D}_l$ with $f_\theta(\cdot)$
 3: calculate $\mathbf{P}_{\mathcal{MD}}$ based on $\hat{\boldsymbol{\Sigma}}^{-1}$
 4: build KNN on $\mathcal{D}_l \cup \mathcal{D}_u$ with $\mathbf{P}_{\mathcal{MD}}$ and $f_\theta(\cdot)$
 5: **for** epoch $\in \{1, 2, \ldots, \text{epoch}_{max}\}$ **do**
 6:     randomly sample $\mathbf{x}_i, \mathbf{x}_j$ from $\mathcal{D}_l \cup \mathcal{D}_u$
 7:     randomly sample $\mathbf{x}_n$ from $\mathcal{B}_k(\mathbf{x}_i)$
 8:     calculate $Loss$ based on Eq.(6)
 9:     optimize $\mathbf{P}$ via SGD according to $Loss$
10: **end for**
11: **return** $f_\theta(\cdot)$ and $\mathbf{P}$

---

**Out-of-distribution Data Set.** We use Tiny ImageNet data set [6] and Large-scale Scene Understanding data set [48] as OOD data sets. The Tiny ImageNet data set (TIN) is a subset of ImageNet, which contains 10,000 test images, includes 200 different classes. Following the settings used by previous studies [31, 43, 49], we use two variants of TIN: TinyImageNet-crop (TINc) and TinyImageNet-resize (TINr), by randomly cropping or downsampling each image to $32 \times 32$, respectively. The Large-scale Scene Understanding data set (LSUN) contains 10,000 testing images belonging to 10 different scene categories. Similarly, we use two variants of LSUN: LSUN-crop (LSUNc) and LSUN-resize (LSUNr). Because some comparison methods in our experiments heavily rely on OOD validation. We randomly draw several images from ID testing images and OOD images as the OOD validation set. The rest of the OOD images are added to unlabeled data $\mathcal{D}_u$ and used as testing data. These OOD data sets are released by ODIN [31] with their code[1].

**Comparsion Methods.** We compare our STEP approach with representative OOD detection methods, including the state-of-the-art UOOD method. ODIN [31] is a common baseline of OOD detection. It uses maximal softmax score combining temperature scaling and input preprocessing tricks to distinguish ID and OOD samples. MAH [27] uses Mahalanobis distance as the OOD confidence score. For features of each layer in the backbone model, it independently calculates the Mahalanobis distances between the target sample and each known class center. Then it integrates them by weighted averaging via an extra OOD validation set. We denote it as MAH [†] because it uses a validation set when training. UOOD [49] utilizes a two-head CNN consisting of one common feature extractor and two classifiers which has different decision boundaries to detect OOD samples. This method optimizes a discrepancy loss between two classifiers during the training stage and uses this discrepancy as the OOD score when testing. However, this method relies on extra OOD validation to perform model selection. Therefore, we denoted it as UOOD [†] in our experiments. For fair comparisons, we also implement a variant of it denoting as UOOD . UOOD that uses discrepancy loss to perform model selection instead of the performance on an extra OOD validation set.

**Evaluation Metrics.** Following the settings used by previous studies [49, 43, 31], we evaluate our approach with five common metrics: AUROC, FPR at $95\%$ TPR, Detection Error, AUPR-In, and AUPR-Out. More details about evaluation metrics are presented in the supplementary material.

**Implementation Details.** In all experiments, we adopt the Densenet-BC [21] as the backbone since it is widely used in previous studies [49, 43, 31]. Our backbone is trained by SOTA contrastive learning method SimCLR [5] for 500 epochs. We set the learning rate to $10^{-3}$ with a cosine annealing strategy. For fair comparisons, each comparison method can use the pre-trained backbone model. MAH [27] uses the features from different layers extracted from the pre-trained backbone model. A well-trained linear classifier with a pre-trained backbone model is provided for ODIN [31] and UOOD [49]. The hyper-parameter K for STEP is set to 12 for all data set pairs. All experiments are performed on one single NVIDIA 3090 graphics card. More details on implementation are provided in the supplementary material and our code has been open source [2].

---

[1]https://github.com/ShiyuLiang/odin-pytorch
[2]https://www.lamda.nju.edu.cn/code_step.ashx

## 3.2 Experiment Results

**OOD Detection Performance.** We evaluate STEP with compared methods on various OOD benchmarks. Analyzed by five common metrics, the results are shown in Tab.(1). From the results, we observe that ODIN suffers from severe performance degradation. Moreover, its performance is close to random guessing in some cases. The limitation of labeled data mainly causes this. We can hardly train a high-quality classification model to provide accurate logits for ODIN . Hence, ODIN can not give the correct judgment based on inaccurate logits. Our STEP approach outperforms methods that do not heavily rely on an OOD validation set by a large margin. Even compared with those methods that heavily rely on the OOD validation set, such as UOOD [†] and MAH [†], our STEP approach is still better than them in most cases. However, a good OOD validation set is expensive and nearly impossible to build in the real world. The number of OOD samples can be infinitely many, and a fixed-size validation set cannot capture the complete OOD information. Therefore, introducing the validation set during training will reduce the model's generalization in the real environment. We will verify this in detail in subsequent experiments.

Table 1: Performance comparison on various OOD benchmarks evaluated by 5 common metrics. Methods with [†] use extra OOD validation set. The best results are indicated in bold. Our approach outperforms other methods in most cases, even though they use an extra OOD validation set.

| Metrics | ID Dataset | OOD Dataset | ODIN | MAH [†] | UOOD | UOOD [†] | STEP |
|---|---|---|---|---|---|---|---|
| AUROC ↑ | Cifar10 | TINc | $81.00 \pm 6.30$ | $87.67 \pm 2.47$ | $90.46 \pm 9.74$ | $99.07 \pm 0.48$ | $\mathbf{99.99 \pm 0.00}$ |
| | | TINr | $59.10 \pm 2.08$ | $86.88 \pm 0.87$ | $84.67 \pm 9.41$ | $92.63 \pm 3.42$ | $\mathbf{95.61 \pm 0.36}$ |
| | | LSUNc | $76.17 \pm 5.37$ | $97.68 \pm 0.09$ | $96.92 \pm 2.04$ | $98.79 \pm 0.67$ | $\mathbf{99.99 \pm 0.00}$ |
| | | LSUNr | $69.05 \pm 3.49$ | $90.41 \pm 1.00$ | $80.87 \pm 24.45$ | $97.81 \pm 0.94$ | $\mathbf{99.07 \pm 0.20}$ |
| | Cifar100 | TINc | $61.65 \pm 6.71$ | $71.15 \pm 2.20$ | $98.34 \pm 1.57$ | $98.84 \pm 0.83$ | $\mathbf{99.99 \pm 0.01}$ |
| | | TINr | $54.46 \pm 0.74$ | $73.94 \pm 1.79$ | $84.80 \pm 8.87$ | $\mathbf{95.31 \pm 0.93}$ | $93.51 \pm 1.17$ |
| | | LSUNc | $46.99 \pm 4.99$ | $93.91 \pm 3.41$ | $97.49 \pm 1.48$ | $99.31 \pm 0.62$ | $\mathbf{99.99 \pm 0.00}$ |
| | | LSUNr | $52.06 \pm 2.24$ | $78.45 \pm 1.11$ | $97.61 \pm 0.55$ | $\mathbf{98.96 \pm 0.40}$ | $98.20 \pm 0.56$ |
| FPR at 95%TPR ↓ | Cifar10 | TINc | $53.37 \pm 10.55$ | $44.17 \pm 6.43$ | $29.35 \pm 30.05$ | $2.75 \pm 1.65$ | $\mathbf{0.00 \pm 0.00}$ |
| | | TINr | $89.76 \pm 1.45$ | $58.57 \pm 3.09$ | $31.72 \pm 11.50$ | $19.61 \pm 9.50$ | $\mathbf{17.63 \pm 1.10}$ |
| | | LSUNc | $64.06 \pm 9.12$ | $7.73 \pm 0.46$ | $6.59 \pm 3.22$ | $3.56 \pm 1.93$ | $\mathbf{0.00 \pm 0.00}$ |
| | | LSUNr | $76.89 \pm 5.04$ | $45.41 \pm 3.87$ | $32.69 \pm 31.93$ | $6.49 \pm 2.89$ | $\mathbf{4.48 \pm 1.02}$ |
| | Cifar100 | TINc | $84.24 \pm 8.02$ | $90.15 \pm 1.99$ | $5.22 \pm 5.59$ | $3.16 \pm 2.25$ | $\mathbf{0.00 \pm 0.01}$ |
| | | TINr | $90.10 \pm 0.46$ | $80.55 \pm 1.89$ | $29.09 \pm 15.68$ | $\mathbf{11.10 \pm 4.21}$ | $23.21 \pm 4.14$ |
| | | LSUNc | $93.49 \pm 2.42$ | $24.93 \pm 21.75$ | $6.24 \pm 3.80$ | $1.93 \pm 2.43$ | $\mathbf{0.00 \pm 0.00}$ |
| | | LSUNr | $89.79 \pm 0.79$ | $69.69 \pm 2.42$ | $4.92 \pm 1.33$ | $\mathbf{2.39 \pm 0.74}$ | $8.25 \pm 3.14$ |
| Detection Error ↓ | Cifar10 | TINc | $25.53 \pm 4.67$ | $19.93 \pm 2.63$ | $11.59 \pm 11.35$ | $2.54 \pm 1.27$ | $\mathbf{0.12 \pm 0.01}$ |
| | | TINr | $43.04 \pm 1.48$ | $20.14 \pm 0.82$ | $18.07 \pm 5.55$ | $11.71 \pm 4.56$ | $\mathbf{10.77 \pm 0.52}$ |
| | | LSUNc | $29.57 \pm 3.82$ | $6.28 \pm 0.25$ | $4.20 \pm 2.12$ | $2.58 \pm 1.32$ | $\mathbf{0.11 \pm 0.01}$ |
| | | LSUNr | $35.52 \pm 2.46$ | $16.23 \pm 0.95$ | $18.40 \pm 15.68$ | $4.99 \pm 1.91$ | $\mathbf{4.66 \pm 0.57}$ |
| | Cifar100 | TINc | $40.95 \pm 5.07$ | $32.58 \pm 1.64$ | $3.67 \pm 3.62$ | $2.76 \pm 1.00$ | $\mathbf{0.32 \pm 0.06}$ |
| | | TINr | $46.36 \pm 0.56$ | $31.09 \pm 1.44$ | $16.53 \pm 7.87$ | $\mathbf{6.88 \pm 2.33}$ | $13.26 \pm 1.61$ |
| | | LSUNc | $48.47 \pm 1.61$ | $11.20 \pm 3.73$ | $4.24 \pm 2.34$ | $2.06 \pm 1.54$ | $\mathbf{0.23 \pm 0.04}$ |
| | | LSUNr | $46.73 \pm 0.66$ | $27.33 \pm 1.03$ | $3.11 \pm 0.78$ | $\mathbf{1.90 \pm 0.51}$ | $6.40 \pm 1.32$ |
| AUPR-In ↑ | Cifar10 | TINc | $76.80 \pm 8.20$ | $85.35 \pm 2.86$ | $89.31 \pm 10.05$ | $98.59 \pm 0.67$ | $\mathbf{99.99 \pm 0.00}$ |
| | | TINr | $57.10 \pm 2.11$ | $86.79 \pm 1.17$ | $79.02 \pm 12.17$ | $88.72 \pm 4.93$ | $\mathbf{94.71 \pm 0.51}$ |
| | | LSUNc | $72.16 \pm 6.60$ | $96.70 \pm 0.21$ | $94.78 \pm 4.07$ | $98.31 \pm 0.92$ | $\mathbf{100.00 \pm 0.00}$ |
| | | LSUNr | $65.37 \pm 3.39$ | $89.93 \pm 1.23$ | $79.41 \pm 19.89$ | $96.86 \pm 1.27$ | $\mathbf{99.02 \pm 0.20}$ |
| | Cifar100 | TINc | $58.29 \pm 5.01$ | $71.18 \pm 2.69$ | $97.55 \pm 2.04$ | $98.24 \pm 1.50$ | $\mathbf{99.99 \pm 0.01}$ |
| | | TINr | $52.96 \pm 0.59$ | $70.95 \pm 2.20$ | $77.32 \pm 9.81$ | $91.67 \pm 1.29$ | $\mathbf{91.91 \pm 1.34}$ |
| | | LSUNc | $47.41 \pm 2.86$ | $92.26 \pm 2.17$ | $95.45 \pm 2.32$ | $99.09 \pm 0.88$ | $\mathbf{99.99 \pm 0.00}$ |
| | | LSUNr | $50.47 \pm 1.75$ | $74.22 \pm 1.14$ | $95.53 \pm 0.95$ | $\mathbf{98.11 \pm 0.78}$ | $98.07 \pm 0.52$ |
| AUPR-Out ↑ | Cifar10 | TINc | $83.63 \pm 5.11$ | $88.67 \pm 2.28$ | $91.34 \pm 8.69$ | $99.32 \pm 0.35$ | $\mathbf{99.99 \pm 0.00}$ |
| | | TINr | $58.83 \pm 1.77$ | $84.26 \pm 0.95$ | $89.21 \pm 6.22$ | $94.60 \pm 2.70$ | $\mathbf{96.31 \pm 0.28}$ |
| | | LSUNc | $78.43 \pm 5.12$ | $98.16 \pm 0.12$ | $98.01 \pm 1.18$ | $99.14 \pm 0.48$ | $\mathbf{99.99 \pm 0.00}$ |
| | | LSUNr | $70.51 \pm 3.97$ | $88.84 \pm 1.20$ | $84.45 \pm 21.48$ | $98.41 \pm 0.70$ | $\mathbf{99.14 \pm 0.19}$ |
| | Cifar100 | TINc | $62.88 \pm 7.90$ | $65.14 \pm 2.21$ | $98.77 \pm 1.23$ | $99.08 \pm 0.51$ | $\mathbf{99.99 \pm 0.01}$ |
| | | TINr | $55.94 \pm 0.71$ | $71.57 \pm 1.71$ | $89.44 \pm 6.96$ | $\mathbf{96.84 \pm 0.82}$ | $94.66 \pm 1.07$ |
| | | LSUNc | $49.91 \pm 4.42$ | $93.77 \pm 5.30$ | $98.33 \pm 0.99$ | $99.39 \pm 0.48$ | $\mathbf{99.99 \pm 0.00}$ |
| | | LSUNr | $55.18 \pm 1.56$ | $78.19 \pm 1.33$ | $98.49 \pm 0.37$ | $\mathbf{99.32 \pm 0.24}$ | $98.35 \pm 0.56$ |

**Generalization of OOD Detection.** Our STEP approach and some previous studies (e.g., UOOD , MAH ) could utilize OOD samples during the training phase. For example, our STEP performs contrastive learning on both ID and OOD data, UOOD optimizes the discrepancy loss on ID and OOD unlabeled data and selects the final model with an extra OOD validation set, MAH tunes their weighting parameters on an OOD validation set. Let known OOD samples denote the OOD samples that the algorithm used during the training phase, contrasting to unknown OOD samples. We want to explore whether the use of known OOD samples will reduce the performance of the model on unknown OOD samples. Therefore, we design a novel experiment for algorithms using OOD samples, in which the model is trained with ID dataset and known OOD samples while tested with unknown samples. As an example, we train the model on the ID data set (CIFAR-10) and OOD data set (TINr) and replace all OOD samples in the test set with a new OOD data set (TINc) when testing. An OOD detection model with strong generalization should obtain consistent performance, no matter what OOD data set we used to construct the testing set. We tested four OOD detection methods on CIFAR-10 with two different OOD data set pairs. From the results shown in Fig.(2), we found that UOOD $^{\dagger}$ and MAH $^{\dagger}$ have severe performance degradation when detecting unknown OOD samples. This phenomenon is because the OOD validation set used by these methods introduces a severe bias to their models. Furthermore, we also conducted experiments to analyze the relationship between OOD detection performance and loss and verified the instability in the training process of the SOTA UOOD $^{\dagger}$ method. We put the extra experiments in the supplementary material. ODIN's performance changes very randomly, which is also in line with expectations because it has only seen ID data during the training process. Our STEP approach gives a high and relatively close performance on both known and unknown OOD data sets, which proves the effectiveness and strong generalization of our approach. Further, we suggest that the experimental method proposed should be verified in all future OOD detection studies that use OOD samples in the training process.

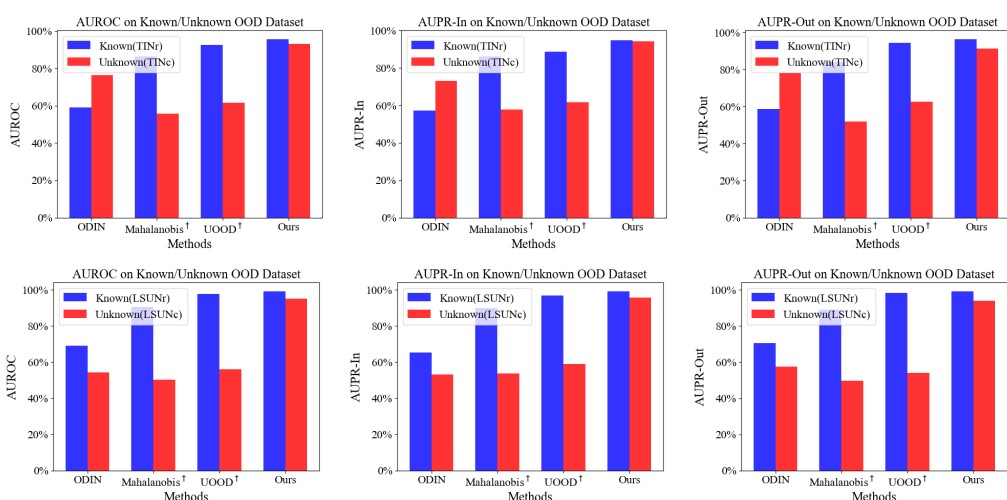

Figure 2: Performance of different methods on Known / Unknown OOD data set evaluated by various metrics. The results shows that our STEP approach not only has very good OOD detection performance, but also can generalize to unknown OOD samples.

**Ablation Study.** As introduced in Section 2, our STEP approach contains four components in total: MAH, KNN, Unzipping, and Structure-Keep. Comprehensive ablation studies are conducted to verify the effectiveness of each component. As shown in Tab.(2), we sequentially add the components of STEP and verify the performance of each model on two OOD benchmarks. The first line in the table shows the results of directly distinguishing the minimum Mahalanobis distance from the target sample to each class center. Since necessary $\hat{\Sigma}$ cannot be accurately estimated, the detection performance is not ideal. The second line proves that the geodetic distance can alleviate the inaccurate estimation problem to a certain extent, thereby improving the detection performance. The third line is the incomplete version of our STEP approach to remove Structure-Keep. The result of this line proves that the Structure-Keep technique is very important. Otherwise, the detection performance will be greatly reduced. The fourth line, our STEP approach, gives the best results. This proves that the four steps proposed in this article can only be integrated together to get the best results.

Table 2: Ablation Study of our STEP approach evaluated by AUROC. This table proves that every part of our approach is indispensable.

| Different parts of STEP | | | | Data set pair | |
|---|---|---|---|---|---|
| MAH | KNN | Unzipping | Sturture-Keep | Cifar10-TINr | Cifar10-LSUNr |
| ✓ | | | | $90.96 \pm 0.28$ | $93.46 \pm 0.51$ |
| ✓ | ✓ | | | $91.26 \pm 1.74$ | $97.35 \pm 0.45$ |
| ✓ | ✓ | ✓ | | $79.58 \pm 0.69$ | $80.38 \pm 0.95$ |
| ✓ | ✓ | ✓ | ✓ | $95.62 \pm 0.39$ | $99.07 \pm 0.20$ |

**Robustness.** In this paragraph, we verify the robustness of STEP to hyper-parameter $K$ and the number of labeled data $|\mathcal{D}_l|$. We test the performance of STEP on different OOD data sets for different choices of K in a large range from 2 to 18. Fig.(3a) shows that STEP is not sensitive to the hyper-parameter K (the number of neighbors when KNN is built). Furthermore, we find that choosing a smaller K helps improve the detection performance. Then we test how the amount of ID labeled data affects the performance of different methods. From the results shown in Fig.(3b), we find that our STEP is very tolerant of the amount of ID labeled data. Even in the case of extremely insufficient ID labeled data, an acceptable performance still can be achieved by our STEP approach.

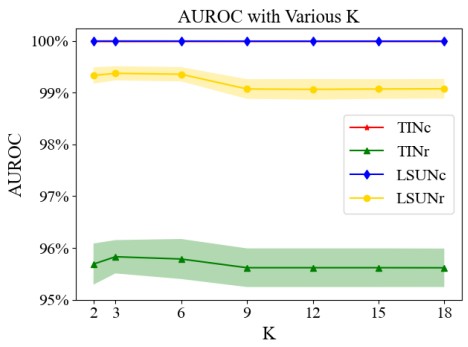 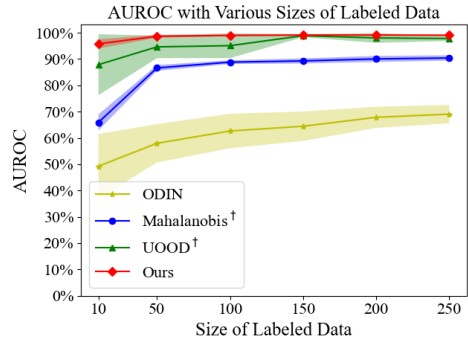

(a) AUROC with various K on different OOD benchmarks.

(b) AUROC of different methods with various sizes of labeled data.

Figure 3: The robustness of STEP approach. (a), (b) show that STEP approach is robust on K and size of labeled data, respectively.

# 4 Related Work

This work is mainly related to self-supervised learning, semi-supervised learning, positive-unlabeled learning, and OOD detection.

**Self-supervised learning.** Self-supervised learning is a powerful framework to learn discriminative feature representations in an unsupervised fashion via artificially designed auxiliary tasks. Recently, contrastive learning [17, 5] shows remarkable progress on it. Benefiting from the progress, some studies [42, 16] utilize the learned representations to cluster samples with unseen labels. STEP proposed in this paper takes advantage of the powerful features derived from the use of contrastive learning. Any progress in comparative learning can be used by STEP to further improve OOD detection performance.

**Semi-supervised learning.** SSL [36] aims to leverage unlabeled data to improve the performance of the model when plenty of labeled data is inconvenient and expensive to access. Our paper is mainly related to deep SSL. The combining of SSL technology and DNNs has significantly improved classification accuracy. Many excellent studies, such as consistency regularization based methods [41, 35], entropy minimization based methods [11] and holistic methods [3, 39], have been proposed in recent years. There are also some studies [13, 50] that focus on improving the safeness of SSL. Specifically, they aim to ensure the performance of SSL when unlabeled data contains OOD samples.

However, these studies all consider the classification performance of the model for known categories under the semi-supervised setting and ignore the problem of overconfidence in the OOD sample when testing. Efforts on this issue remain limited. Therefore, we propose the semi-supervised OOD detection setting and design STEP approach for it.

**Positive-unlabeled learning.** PU learning [1] is the setting where a learner only has access to positive examples and unlabeled data. Studies in this direction can be mainly divided into three categories: two-step techniques [32], biased learning [33] and class prior incorporation [7]. Some recent studies expand this technique into the situation that includes anomalies and OOD samples. ADOA [51] considers the anomaly detection problem where we only can observe some labeled anomalies along with unlabeled data. PUC [46] aims to select data for network compression from massive unlabeled data that may contain OOD samples. However, these methods only consider the known distributions which pay little attention on the detection performance of unknown distribution.

**OOD detection.** OOD detection has been studied for a long history. The baseline [18] of this problem attempts to detect OOD samples depending on the predicted softmax class probability. Modified generative adversarial networks [26] are used to generate challenging OOD samples during the training stage, and the algorithm encourages the classifier to assign OOD samples uniform class probabilities. ODIN [31] applies the temperature scaling and input preprocessing to further strengthen the difference between ID samples and OOD samples. ELOC [43] uses the ensemble of K leave-out classifiers to detect OOD samples. There are some other studies that use energy-based models [34, 12], hierarchical relations [23, 24], and so on. The current state-of-the-art method [49] for OOD detection utilizes the discrepancy between two classifiers to separate ID and OOD samples. Nevertheless, these studies either assume that there is an accurate classification model or assume that there is sufficient labeled data, which limits their application in the real world. There are also some unsupervised OOD detection studies [40, 19, 2, 38] utilizing the power of the contrastive learning framework. However, although these studies do not require labels, they still need a large amount of ID data for training. Previous studies [13] have reported that collecting clean unlabeled data is also very difficult in the real world. Hence, we study a more general setting that is very common in real-world applications in this paper.

# 5 Conclusions

In this paper, we propose a novel OOD detection setting, called semi-supervised OOD detection. In this setting, we aim to distinguish ID and OOD samples by using limited ID labeled data and large amounts of mixed unlabeled data. Due to the generality of this setting, it commonly occurs in real-world applications. In the case of only having limited ID labeled data, we find that the previous studies have suffered performance degeneration, mainly due to the inaccurate estimation from the limited ID data. Focusing on this setting, we propose a novel STEP approach. Our main idea is to detect OOD samples in a detection-specific space where we maintain the same local topological structures as the original feature space. Our STEP approach outperforms other methods by a large margin in most cases and achieves remarkable detection performance on several benchmarks. Meanwhile, we also conduct comprehensive experiments to verify the robustness and generalization of our STEP approach. The limitation of our work is the lack of solid theoretical results. Broadly speaking, other OOD detection methods also have similar problems. We will put efforts into the theoretical understanding of OOD detection in future work.

# Broader Impact

In this work, we study OOD detection, which is a fundamental problem in deep learning. Specifically, we first proposed a novel OOD detection setting. In this setting, only limited ID labeled data and many mixed unlabeled data can be used for OOD detection. This is a novel and practical setting commonly appearing in real-world applications because under this setting, we neither require a large amount of labeled data nor clean unlabeled data. We propose the STEP approach to detect OOD samples in a detection-specific space, greatly improving the performance of OOD detection. Our work will give instructions for those applications having difficulties collecting large quantities of pure ID labeled data while demanding detecting OOD samples to prevent potential dangers in real-world applications. At the same time, there is still much room for exploration in this setting. We hope our

work can inspire more discussions about OOD detection in real scenarios and drive more researchers to build practical and robust OOD detection algorithms.

Meanwhile, we are aware that abuse of this technology can pose ethical issues. In particular, we note that people expect that real people rather than algorithms make the judgments behind the system. Despite the risks of such AI research, developing and demonstrating such technologies is essential to understand the technology's practical and potentially troublesome applications. We hope that the responsible use of technology will stimulate discussion about these methods' practices and controls.

## Acknowledgment

This research was supported by the NSFC (62176118, 61921006), and the Hikvision Cooperation Fund.

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
