# Supplementary for "STEP : Out-of-Distribution Detection in the Presence of Limited In-distribution Labeled Data"

**Zhi Zhou[1]\***, **Lan-Zhe Guo[1]\***, **Zhanzhan Cheng[2]**, **Yu-Feng Li[1]†**, **Shiliang Pu[2]**

[1]National Key Laboratory for Novel Software Technology, Nanjing University, Nanjing, China
`{zhouz, guolz, liyf}@lamda.nju.edu.cn`
[2]Hikvision Research Institute, Hangzhou, China
`{chengzhanzhan, pushiliang.hri}@hikvision.com`

## A  Appendix

### A.1  Evaluation Metrics

In this paper, we use the following five metrics, which are used by previous studies [6, 5, 2] to evaluate the ability of our STEP approach to distinguish the in- and out-of-distribution data. We denote $\text{TP}, \text{TN}, \text{FP}, \text{FN}$ as true positives, true negatives, false positives, and false negatives, respectively.

**AUROC** is the Area Under the Receiver Operating Characteristic curve [1], which depicts the relationship between $\text{TPR} = \frac{\text{TP}}{\text{TP}+\text{FN}}$ and $\text{TPR} = \frac{\text{FP}}{\text{FP}+\text{TN}}$ over all possible score thresholds. This metric can be interpreted as the ability that the model assigns a positive sample a higher detection score than a negative example.

**FPR at 95%TPR** measures the probability that a negative (out-of-distribution) sample is misclassified as a positive (in-distribution) sample when the TPR is up to $95\%$. This metric evaluates the method's precision when recall most of the out-of-distribution samples.

**Detection Error** measures the minimum misclassification probability, which is calculated by minimum average TPR and FPR overall score thresholds. Here, we assume that both positive and negative examples have an equal probability of appearing in the test set.

**AUPR-In** is the Area under the Precision-Recall curve[3] which is a graph showing the $\text{precision} = \frac{\text{TP}}{\text{TP}+\text{FP}}$ and $\text{recall} = \frac{\text{TP}}{\text{TP}+\text{FN}}$ against each other. Here, the precision-recall curve treats in-distribution samples are specified as positives.

**AUPR-Out** is similar to AUPR-In. The only difference is that the precision-recall curve treats out-of-distribution samples as positives.

### A.2  Implementation Details

All experiments were repeated five times with the random seed setting from 0 to 4. For SimCLR, We adopt the version[1] implemented by Pytorch to train our backbone network. We train our backbone network for 500 epochs with 512 images in a batch. The feature dimension used in SimCLR is set to 512. Other parameters are the same as the default settings. For extracting local structure, we adopt Faiss[2] to build a KNN graph with K set to 12. For our STEP approach, we train $\mathbf{P}$ for 1500 epochs with 512 image tuples in a batch. The learning rate and weight decay are set to 0.0003 and

---

*Contribute to this work equally

†Corresponding author

[1]https://github.com/sthalles/SimCLR

[2]https://github.com/facebookresearch/faiss

35th Conference on Neural Information Processing Systems (NeurIPS 2021).

1e-4, respectively. For ensembling of latent representations, flatten the representation from 1st cov and 3 blocks in the densenet and concat them as the representation for methods that require latent representations. Refer to our code implementation for more details.

## A.3 Stability of Training

We track the relationship between loss and performance on the validation set for UOOD [†] and STEP during the training on several OOD detection benchmarks. The results are shown in Fig.( 1). The images in the first row show the loss and AUROC per epoch of our STEP approach and the images in the second row show the same thing for the SOTA UOOD method. The images in the same column show the evaluation of different methods on the same OOD dataset. The results show that, for our STEP approach, the AUROC keeps increasing while the loss is decreasing on the big trend during the training stage. However, the same thing does hold for UOOD . The AUROC changes a lot in a wide range while loss decreases for UOOD in the training stage. Therefore, UOOD's training is not stable enough, and it needs an extra OOD validation set to perform model selection.

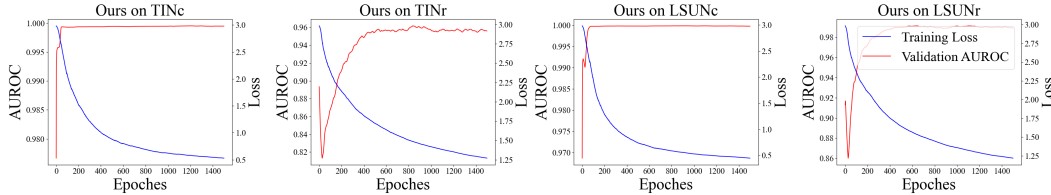

(a) Our STEP approach's loss and AUROC on different OOD benchmarks.

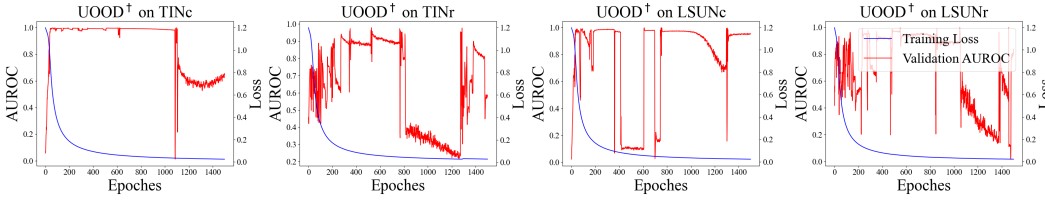

(b) UOOD 's loss and AUROC on different OOD benchmarks.

Figure 1: (a) shows our STEP approach's loss and AUROC on an OOD validation set for each epoch during the training. (b) shows the same thing for the SOTA UOOD method. Results show that the AUROC increases while loss decreases for STEP . However, this does not hold on UOOD .

## A.4 Generalization Changes

We further test generalization on the extensive OOD data sets and add comparison with an unsupervised OOD detection method [4]. For the fair comparison, SSD uses the same pre-trained network as our STEP . The results in Tab.(1) show that SSD and ODIN are affected by the known OOD samples in the training data, resulting in poor detection performance on the known OOD samples. Relatively, their detection performance on unknown OOD samples is improved. Consistent with the previous conclusion, when MAH and UOOD face unknown OOD samples, there will be great detection performance fluctuations. Only our STEP approach can achieve an accurate and relatively stable detection performance on both known and unknown OOD samples.

Table 1: Performance of different methods on Known / Unknown OOD data set evaluated by AUROC. Our proposal achieves accurate and stable detection performance. The ID data set is CIFAR-10.

|  | Known OOD samples | Unknown OOD samples | | | | | Avg. Performance |
|---|---|---|---|---|---|---|---|
|  | TINr | TINc | LSUNr | LSUNc | iSUN | |
| ODIN | $59.25 \pm 2.01$ | $76.19 \pm 2.73$ | $66.36 \pm 5.12$ | $60.82 \pm 2.44$ | $64.51 \pm 4.04$ | 65.43 |
| MAH | $75.20 \pm 1.97$ | $55.67 \pm 6.30$ | $81.07 \pm 1.44$ | $17.74 \pm 8.34$ | $74.40 \pm 1.70$ | 60.82 |
| UOOD | $94.44 \pm 2.05$ | $69.78 \pm 4.24$ | $98.39 \pm 1.13$ | $57.20 \pm 10.39$ | $96.74 \pm 2.04$ | 83.31 |
| SSD | $68.34 \pm 1.12$ | $83.85 \pm 1.93$ | $67.92 \pm 1.62$ | $94.62 \pm 0.66$ | $70.00 \pm 1.40$ | 76.95 |
| STEP | $95.62 \pm 0.37$ | $93.19 \pm 1.43$ | $98.29 \pm 0.21$ | $91.67 \pm 1.91$ | $97.42 \pm 0.36$ | 95.23 |

## A.5 Varying the OOD Ratio

Since the ratio of OOD samples may affect the feature representations which play an important role in our approach, we conduct an extra experiment to study the impact of different OOD ratios in unlabeled data. Due to limited size of benchmark data sets, we fix unlabeled data set with 10,000 samples and try five different OOD ratios $\gamma \in \{10\%, 25\%, 50\%, 75\%, 90\%\}$, where $\gamma$ denotes $1 - \frac{\sum_{\boldsymbol{x} \in \mathcal{D}_u} \mathbb{1}(g(\boldsymbol{x}) \in \mathcal{C}_l)}{|\mathcal{D}_u|}$ and $g(\cdot)$ can obtain the ground-truth label of one sample. Fig.(2) shows the results on four OOD data sets, where the ID data set is CIFAR-10. Comparing these results, we will find that the performance of our proposal remains relatively stable at different OOD ratios. When the task is complex and the OOD ratio is extreme, the performance of our proposal will decrease. In addition, the reduction of unlabeled data also leads to deterioration of detection performance. This matters little because unlabeled data is cheap and easy to obtain in large quantities in practical applications. In these cases, how to ensure the OOD detection performance will be a proposition to explore.

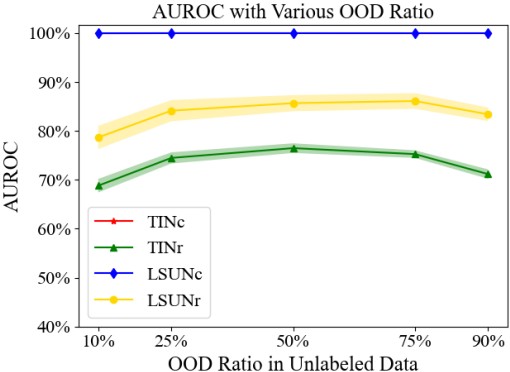

Figure 2: Performance of different OOD ratio in unlabeled data.

## A.6 Visualization

We test the ability to use $\mathcal{MD}$ to distinguish OOD samples when $\hat{\boldsymbol{\Sigma}}$ is estimated with different scales of errors. We use the Equ.(5) to convert the Mahalanobis distance into the Euclidean distance calculation in the projection space. And then use the dimensionality reduction technique in the projection space to get the three pictures on the left. We visualize samples in the detection space of STEP , and get the rightmost picture. The Fig.(3) shows that STEP not only solves the inaccurate estimation problem but also distinguishes ID and OOD samples by a more significant margin.

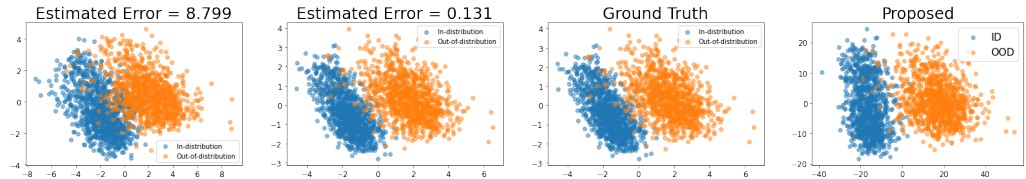

Figure 3: Visualize STEP and Mahalanobis distance's ability to distinguish between OOD and ID samples through dimensionality reduction. Compared with Mahalanobis distance when we estimate unbiased $\hat{\boldsymbol{\Sigma}}$, our STEP approach separates ID and OOD samples by a more significant margin.