# OpenReview forum: "STEP: Out-of-Distribution Detection in the Presence of Limited In-Distribution Labeled Data"
_NeurIPS.cc/2021/Conference — NeurIPS 2021 Poster_

### Official Review · Reviewer_U8Ns · 2021-06-24

**Rating:** 3
**Confidence:** 4

**Summary:**

This paper studies semi-supervised learning with limited label data and  unlabeled data with OOD samples. The covariance in the Mahalanobis distance cannot be estimated in a robust way when only limited labeled data is available. The main idea of the paper is to find a projection  space that will preserve the local topology of the data points. More specifically, the projection matrix should maximize the distance between a given sample and all other  samples subject to the condition that the Mahalanobis distances between this point and points within its local neighborhood are preserved.

**Ethical Concerns:**

None foreseen.

**Limitations And Societal Impact:**

Yes.

**Main Review:**

If Mahalanobis distance is not a reliable measure due to limited data size then it is not clear why the proposed approach is trying to preserve this distance in the objective function.

The Mahalanobis distance in the original space becomes the Euclidean distance in the projected space if the projection matrix is selected as the lower triangular matrix of the precision matrix. OK but how is this going to help estimate a more robust precision matrix? In order for Cholesky to work the precision matrix has to be well-conditioned, which means for every class at least d+1 samples are needed.

Not sure why the OOD datasets are selected from different datasets when a subset of the classes in CIFAR-10  or CIFAR-100 could have been used as OOD. Using scene dataset as OOD in CIFAR-10/100 makes the problem unrealistically easier.

Writing of the paper needs improvement. There were many grammar issues and broken sentences.

---------------
Update:

Regrettably, the author rebuttal does not address my main concerns about the paper. I am not convinced that the main idea of the paper that aims to leverage ill-conditioned Mahalanobis distance to learn useful structural information about the data so as to more effectively perform OOD detection will be very effective in OOD detection.  I stand with my original rating of "clear reject".


**Time Spent Reviewing:**

2 hours

---

> ### Author Response · Authors · 2021-08-09
> **Reply to Reviewer U8Ns**
>
> We sincerely thank you for your time and efforts.
>
> > Q1&2: If Mahalanobis distance is not a reliable measure due to limited data size then it is not clear why the proposed approach is trying to preserve this distance in the objective function.
> The Mahalanobis distance in the original space becomes the Euclidean distance in the projected space if the projection matrix is selected as the lower triangular matrix of the precision matrix. OK but how is this going to help estimate a more robust precision matrix? In order for Cholesky to work the precision matrix has to be well-conditioned, which means for every class at least d+1 samples are needed.
>
> As shown in Table 2, the inaccurate estimation of Mahalanobis distance leads to the degeneration of OOD detection performance. Exploiting the imperfect Mahalanobis distance is of practical importance. The main idea of our paper is to learn a reliable OOD detection measure by using the exploiting the imperfect Mahalanobis distance and the helpful local structure between samples.
>
> > Q3: Not sure why the OOD datasets are selected from different datasets when a subset of the classes in CIFAR-10 or CIFAR-100 could have been used as OOD. Using scene dataset as OOD in CIFAR-10/100 makes the problem unrealistically easier.
>
> Following the previous studies [29, 43], we selected the same data sets to verify the effectiveness of our approach.

---

### Official Review · Reviewer_TQrA · 2021-06-25

**Rating:** 5
**Confidence:** 4

**Summary:**

This paper proposed semi-supervised OOD detection and a novel objective formulation that learns a new representation space in which OOD samples could be separated well.

**Limitations And Societal Impact:**

Yes

**Main Review:**

Pros: The setting is reasonable and the solution seems valid.

Cons:
1.	From my part, this work seems to heavily rely on pre-trained networks by contrastive learning. I am curious to see how is the method perform compared to the pure contrastive learning-based method, such as CSI [1] and SSD[2]. Unfortunately, Table 2 contains no information about this comparison.
2.	The algorithm seems to have a O(n^2) complexity, I am excited to know what is the typical training time on benchmark datasets.
3.	It looks like the hyperparameters are tuned on the test set in Figure 3 and the best result is presented in the main table. Besides, I wonder how is the weight determined for the two losses? Is the weight 1 optimal one?
4.	It would be better to present the results under some other detection metrics in the experiment section, such as the Mahalanobis distance, energy score.
5.	I am wondering how is your method compared with those baselines based on likelihood ratios under deep generative models? Moreover, some popular baselines are important to compare with from my side, such as MSP, Gram matrices[3].
6.	It is a bit unclear that how is the ensemble of representations from each backbone network’s layer used in experiments? Does each baseline have such a design?
7.	It is interesting to look at the in-distribution classification accuracy of your proposed method on benchmark datasets.

Minor points:
1.	Maybe it will give the proposed work a more comprehensive evaluation if some results on different network architectures are presented, say Wideresnet?
2.	It is interesting to observe whether semi-supervised methods, such Mixmatch[4] perform well for OOD detection?

[1]Jihoon Tack, Sangwoo Mo, Jongheon Jeong, and Jinwoo Shin. Csi: Novelty detection via contrastive learning on distributionally shifted instances. In Advances in Neural Information Processing Systems, pages 11839–11852, 2020

[2]Vikash Sehwag, Mung Chiang, and Prateek Mittal. Ssd: A unified framework for self-supervised outlier detection. In Proceedings of the 9th International Conference on Learning Representations, pages 1–17,2021

[3]Chandramouli Shama Sastry, Sageev Oore, Detecting Out-of-Distribution Examples with In-distribution Examples and Gram Matrices, ICML2020

[4]David Berthelot, Nicholas Carlini, Ian Goodfellow, Nicolas Papernot, Avital Oliver, Colin Raffel, MixMatch: A Holistic Approach to Semi-Supervised Learning, NeurIPS 2019



**Time Spent Reviewing:**

10 hours

---

> ### Author Response · Authors · 2021-08-09
> **Reply to Reviewer TQrA**
>
> We thank the reviewer for your detailed and valuable feedback. We mainly detail the response to the main concerns.
>
> > Cons 1: From my part, this work seems to heavily rely on pre-trained networks by contrastive learning. I am curious to see how is the method perform compared to the pure contrastive learning-based method, such as CSI and SSD. Unfortunately, Table 2 contains no information about this comparison.
>
> According to the results of our experiments and previous studies [36], the features obtained by unsupervised learning do have a certain OOD detection ability, which is also a guarantee for the success of our algorithm.
> Taking CSI as an example, although there is no need for the labels, the unlabeled data used must be drawn from in-distribution. This provides the possibility to estimate in-distribution from all unlabeled data. However, in our setting, unlabeled data is mixed with both ID samples and OOD samples that brings difficulties to the application of such methods.
>
> > Cons 2: The algorithm seems to have a O(n^2) complexity, I am excited to know what is the typical training time on benchmark datasets.
>
> Since this O(n^2) part can be accelerated using a good toolkit (e.g., faiss), it is usually taking only a few seconds on our benchmark data sets. (In the case of a data scale of 60,000, this part only takes 6.4 seconds.) And it's worth noting that this part only needs to be calculated once during training.
>
> > Cons 3: It looks like the hyperparameters are tuned on the test set in Figure 3 and the best result is presented in the main table. Besides, I wonder how is the weight determined for the two losses? Is the weight 1 optimal one?
>
> Figure 3 is an experiment on the robustness of hyper-parameters. The hyper-parameters of the main table are mentioned in Section A.2. Since our STEP approach is robust for hyper-parameters, all results in the main table are obtained by K=12 (it's not the best one shown in Figure 3), and setting the weight to 1 always works.
>
> > Cons 4: It would be better to present the results under some other detection metrics in the experiment section, such as the Mahalanobis distance, energy score.
>
> In fact, the purpose of this paper is to learn a new metric to detect OOD samples. Calculating Mahalanobis distance in the learned representation space will still have the problem of inaccurate estimation. Calculating energy score requires a classifier, which requires an SSL algorithm that can handle OOD samples, which is beyond the scope of this paper.
>
> > Cons 5: I am wondering how is your method compared with those baselines based on likelihood ratios under deep generative models? Moreover, some popular baselines are important to compare with from my side, such as MSP, Gram matrices.
>
> We compare SOTA method UOOD, which is more consistent with the problem setting in our paper. These experiments can prove the effectiveness of our method. The SSL baseline proposed by the Review 7FHe is an issue worth studying and we posted the related results.
>
> > Cons 6: It is a bit unclear that how is the ensemble of representations from each backbone network’s layer used in experiments? Does each baseline have such a design?
>
> We thank the reviewer for pointing out this issue. We have mentioned that our approach uses an ensemble of latent representations in L.125-126. We flatten the representation from 1st cov and 3 blocks in densenet and concat them as the representation for our STEP approach. We will make this pointer clearer in the final version. The comparison method using latent representations also uses this design.
>
> > Cons 7: It is interesting to look at the in-distribution classification accuracy of your proposed method on benchmark datasets.
>
> We must emphasize that our work is still focused on the performance of OOD detection. After accurately distinguishing OOD samples, a classification model with better performance can be obtained by using semi-supervised or supervised learning.
>
> > Minor points 1: Maybe it will give the proposed work a more comprehensive evaluation if some results on different network architectures are presented, say Wideresnet?
> We experimented with WideResNet-28-2 on the (Cifar10, TINc) benchmark and got similar results.
>
> |                 | ODIN              | MAH*               | UOOD              | Proposed         |
> | --------------- | ----------------- | ------------------ | ----------------- | ---------------- |
> | AUROC           | 68.33 $\pm$ 15.50 | 77.86 $\pm$ 3.76   | 87.27 $\pm$ 17.24 | 99.20 $\pm$ 0.26 |
> | TPR95           | 58.14 $\pm$ 18.19 | 69.49 $\pm$ 5.39   | 20.73 $\pm$ 25.00 | 2.38 $\pm$ 1.28  |
> | Detection Error | 30.83 $\pm$ 9.74  | 28.83 $\pm$ 3.32   | 11.56 $\pm$ 13.06 | 3.56 $\pm$ 0.60  |
> | AUPR-In         | 61.34 $\pm$ 13.56 | 75.91 $\pm$ 4.03   | 82.86 $\pm$ 19.46 | 99.35 $\pm$ 0.20 |
> | AUPR-Out        | 75.51 $\pm$ 12.73 | & 77.70 $\pm$ 3.70 | 91.14 $\pm$ 12.46 | 99.04 $\pm$ 0.33 |
>
> > Minor points 2: It is interesting to observe whether semi-supervised methods, such Mixmatch[4] perform well for OOD detection?
>
> The SSL baseline proposed by the Review 7FHe is an issue worth studying and we posted the related results.

---

### Official Review · Reviewer_Jitz · 2021-07-16

**Rating:** 3
**Confidence:** 5

**Summary:**

The paper introduces a semi-supervised OOD detection setting and proposes STEP as an approach for this situation. This presented methods combines a Mahalanobis distance with a semi-supervisedly trained backbone model using topological structure preserving optimized projections.


**Limitations And Societal Impact:**

Yes, the authors have included a very adequate Broader Impact section.

**Main Review:**

The description of the methodology is not clear enough for me to see that it is sound and that the motivating principles are fulfilled.
The presented evaluation is insufficient to judge the efficacy of the presented method. The results are only shown for four OOD datasets, with two pairs of them being the same data only downscaled in different ways.
Availability of validation data from the tested out-distribution can not be assumed in an OOD detection setting, as is mentitoned in l.200. However, the evaluation (Fig. 2) for this most relevant setting of 'new OOD datasets' is extremely lacking: only two evaluations are shown, and for each the OOD dataset used for training only differs from the test dataset in in image rescaling methods. A sufficient evaluation would need to include several new OOD datasets, for example as in https://arxiv.org/abs/1812.04606 (Hendrycks et al.'s Outlier Exposure) Table 7.
Overall, problems of language and structure make the paper very difficult to read and particularly the methodological section hard to assess. The results section is not comprehensive enough to appraise the practical benefits of the method.


Questions and points that remained unclear to me:
 - l. 96 How is the covariance matrix calculated? It is taken over all classes, and estimated on all ID samples, right? If so, how do missing labels make it less accurate?
 - Eq. 3: Should the first line be a sum instead of a quantifier over all i,j?
 - The discussion in 2.2 works with Mahalanobis distance only in the feature space (=input image space). Previous approaches [27] use latent representations of a classifier. Could this not be applied here with a classifier trained on the limited labeled data? In 2.3.2. it is still not clear to me where intermediate representations are actually used, since a backbone is mentioned for the calculation of Sigma-hat, but it is still a matrix in image space.
 - l. 142 & Eq. 7: What are the "detection-specific space" and N?

**Time Spent Reviewing:**

5

---

> ### Author Response · Authors · 2021-08-09
> **Reply to Reviewer Jitz**
>
> We follow the experimental settings that were widely used in previous studies [43][29].
>
> > Q: l. 96 How is the covariance matrix calculated? It is taken over all classes, and estimated on all ID samples, right? If so, how do missing labels make it less accurate?
>
> Traditionally, the covariance matrix is calculated on all classes from in-distribution. In our setting, since the unlabeled data is mixed with both ID and OOD samples, the optimal covariance matrix cannot be obtained by directly using all unlabeled data. For this reason, we propose a new technique: Structure-Keep Unzipping, which
> additionally uses local structures between samples to achieve better OOD detection performance.
>
> > Q: Eq. 3: Should the first line be a sum instead of a quantifier over all i,j?
>
> We will revise it throughout the paper.
>
> > Q: The discussion in 2.2 works with Mahalanobis distance only in the feature space (=input image space). Previous approaches use latent representations of a classifier. Could this not be applied here with a classifier trained on the limited labeled data? In 2.3.2. it is still not clear to me where intermediate representations are actually used, since a backbone is mentioned for the calculation of Sigma-hat, but it is still a matrix in image space.
>
> Our approach uses an ensemble of latent representations in L.125-126. We flatten the representation from 1st cov and 3 blocks in the densenet and concat them as the representation for our approach. In addition, the comparison method MAH does use latent representations in our experiments according to the technical details in [27].
>
> > Q: l. 142 & Eq. 7: What are the "detection-specific space" and N?
>
> The detection-specific space is defined in L.121-122 and N is defined in Eq.(7).

---

### Official Review · Reviewer_7FHe · 2021-07-18

**Rating:** 6
**Confidence:** 4

**Summary:**

The paper attempts to extend the existing OOD detection problem to a real world scenario where the in-distribution data is only partially labeled (semi-supervised learning), unlike existing OOD problem formulation where they assume that in-distribution data is fully labeled. The proposed approach, termed as STEP in the paper, extends the Mahalanobis distance based OOD approach into the given semi-supervised OOD detection problem formulation. The proposed approach is evaluated on OOD benchmark with the most recent methods proposed for addressing traditional OOD formulation. Extensive experiment shows the effectiveness of proposed method for the proposed semi-supervised OOD detection problem as compared to existing methods.

**Limitations And Societal Impact:**

- Though the problem formulation is novel and clearly explained, there is still an inherent assumption over the data which is not given as much attention. There is an inherent assumption in the proposed semi-supervised OOD formulation that all the unlabeled in-distribution data belongs to only same K category as that of partially labeled data. However, if we were to consider that majority of training data is unlabeled in a true sense, there is no way of knowing what categories are present. If we consider this case, the proposed formulation should be modified to include known-unknown-distribution in unlabeled data as well, in addition to unknown-unknown-distribution used only during testing.


- A possible way to change this formulation in my suggestion is borrowing few aspects of open-set semi supervised learning problem formulation as proposed here [1]. Though, it only deals with known-unknown-distribution in unlabeled data, in my opinion, it is more realistic scenario and stays true to the practical definition of unlabeled data.


- The paper does an excellent job in comparing with existing most recent methods proposed to solve supervised OOD detection. However, the need of the proposed method is not properly established through experiments. To fully validate why we need to follow the proposed method for semi-supervised OOD, a baseline having SSL method as first step and OOD detector as second step should be created, SSL+OOD. For example, the model should be trained with a popular SSL method first like MixMatch [2] and later the model should utilize confident predictions to create Mahanalobis based [3] OOD detector. This provides a strong baseline for proposed problem formulation. Since, it the method compared in the paper were created with the supervised OOD detection in mind, directly comparing with them is not entirely fair to those methods. The improvement over suggested baseline would provide a more accurate comparison as it is trained by considering both SSL and OOD nature of the problem individually. In my opinion, the effectiveness of the method should come from the consideration of SSL and OOD nature jointly while developing the method.



[1] Yu, Qing, et al. "Multi-task curriculum framework for open-set semi-supervised learning." European Conference on Computer Vision. Springer, Cham, 2020.


[2] Berthelot, David, et al. "MixMatch: A Holistic Approach to Semi-Supervised Learning." Advances in Neural Information Processing Systems 32 (2019).


[3] Lee, Kimin, et al. "A simple unified framework for detecting out-of-distribution samples and adversarial attacks." Advances in neural information processing systems 31 (2018).

**Main Review:**

+ The paper is very well written, has good structure, and is easy to follow. It does a good job in proposing the problem formulation and clearly explaining how the proposed OOD detection problem formulation defers from the typically used OOD detection problem formulation.


+ The paper focuses on Mahanalobis distance based OOD detection. Sec.2.2 points out some of the difficulties in adapting existing Mahanalobis distance based OOD detection methods to the proposed semi-supervised OOD setting. The arguments made by the paper are reasonable and the formulation for semi-supervised OOD that follows also makes sense.


+ Based on the objective formulated to overcome the estimation inaccuracies in computing covariance matrix under semi-supervised OOD formulation. To create a robust Mahanalobis distance based OOD detector, a novel training strategy is derived from the updated formulation which utilizes a projection matrix to overcome on the limited availability of labeled data.


+ Experiments are conducted on standard benchmark datasets used for supervised OOD formulation. The comparison with most recent OOD methods is also provided and across multiple metrics, the proposed method is able to achieve better performance. Specifically, over the regular Mahanalobis method. Furthermore, ablation analysis shows different configuration of the proposed method providing key insights into the various aspects of the proposed idea.





**Time Spent Reviewing:**

3

---

> ### Author Response · Authors · 2021-08-09
> **Reply to Reviewer 7FHe**
>
> Thank you for the valuable suggestions. Please refer to our reply below.
>
> > Q: Though the problem formulation is novel and clearly explained, there is still an inherent assumption over the data which is not given as much attention. There is an inherent assumption in the proposed semi-supervised OOD formulation that all the unlabeled in-distribution data belongs to only same K category as that of partially labeled data. However, if we were to consider that majority of training data is unlabeled in a true sense, there is no way of knowing what categories are present. If we consider this case, the proposed formulation should be modified to include known-unknown-distribution in unlabeled data as well, in addition to unknown-unknown-distribution used only during testing.
> A possible way to change this formulation in my suggestion is borrowing few aspects of open-set semi supervised learning problem formulation as proposed here [1]. Though, it only deals with known-unknown-distribution in unlabeled data, in my opinion, it is more realistic scenario and stays true to the practical definition of unlabeled data.
>
> The scenario you present is a more difficult setting, introducing the imbalance problem of ID classes (missing one of ID classes can be considered as an extreme imbalance problem) and OOD problems into unlabeled data based on the traditional semi-supervised setting. Studying such a valuable and realistic setting has not been well studied yet and is a follow-up direction of our work.
>
> > Q: The paper does an excellent job in comparing with existing most recent methods proposed to solve supervised OOD detection. However, the need of the proposed method is not properly established through experiments. To fully validate why we need to follow the proposed method for semi-supervised OOD, a baseline having SSL method as first step and OOD detector as second step should be created, SSL+OOD. For example, the model should be trained with a popular SSL method first like MixMatch [2] and later the model should utilize confident predictions to create Mahanalobis based [3] OOD detector. This provides a strong baseline for proposed problem formulation. Since, it the method compared in the paper were created with the supervised OOD detection in mind, directly comparing with them is not entirely fair to those methods. The improvement over suggested baseline would provide a more accurate comparison as it is trained by considering both SSL and OOD nature of the problem individually. In my opinion, the effectiveness of the method should come from the consideration of SSL and OOD nature jointly while developing the method.
>
> We use the popular SSL algorithm FixMatch to obtain a classification model on (Cifar10-TINc) benchmark (Top-1 accuracy on ID testing set is 93.47%). Based on this model, ODIN and Mahanalobis are used for OOD detection.
>
> | (Cifar10-TINc)  | ODIN   | MAH    |
> | --------------- | ------ | ------ |
> | AUROC           | 7.35%  | 69.56% |
> | TPR95           | 99.87% | 78.36% |
> | AUPR-In         | 31.50% | 66.20% |
> | AUPR-Out        | 31.42% | 69.92% |
> | Detection Error | 50%    | 35.36% |
>
> The performance of the logit-based OOD detection algorithm (e.g., ODIN) is poor because the OOD samples in the unlabeled data seriously affect the classifier. The performance of the representation-based method (e.g., Mahanalobis) decreases slightly, which shows that the learned feature representation is also affected by the OOD samples in unlabeled data. In summary, simply using a combination of SSL and OOD detection as a baseline does not work well.

---

### Decision · Program_Chairs · 2021-09-28

**Decision:**

Accept (Poster)

**Comment:**

The paper attempts to extend the existing OOD detection problem to a real world scenario where the in-distribution data is only partially labeled (semi-supervised learning), unlike existing OOD problem formulation where they assume that in-distribution data is fully labeled. The proposed approach, termed as STEP in the paper, extends the Mahalanobis distance based OOD approach into the given semi-supervised OOD detection problem formulation. The proposed approach is evaluated on OOD benchmark with the most recent methods proposed for addressing traditional OOD formulation. Extensive experiment shows the effectiveness of proposed method for the proposed semi-supervised OOD detection problem as compared to existing methods.

However, there exists some limitations as follows.

1) Theory: the theory part is not thoroughly evaluated.

2) Experiments: 2.1) Insufficient OOD experiments: the authors only consider CIFAR-10/100, while most OOD papers also experiment with SVHN and TinyImageNet dataset. 2.2) Baselines: how is your method compared with those baselines based on likelihood ratios under deep generative models? Moreover, some popular baselines are important to compare with, such as MSP, Gram matrices.

In the end, although the idea and problem formulation presented in the paper has sufficient contribution to make to the field, I think that this paper may not be ready for publication at NeurIPS, but the next version must be a strong paper if above limitations can be well addressed.

**Consistency Experiment:**

NeurIPS has a long history of experimentation. In 2014, NeurIPS ran an experiment in which 10% of submissions were reviewed by two independent committees to quantify the randomness in the review process. This year, we repeated a variant of this experiment to see how the quality of the review process has changed over time.  This paper was part of the experiment and was therefore assigned to two committees (consisting of reviewers, an Area Chair, and a Senior Area Chair) that reached independent decisions.  If both committees made the same recommendation, this recommendation was followed. If a single committee recommended acceptance, the paper was accepted (with the exception of a few cases in which the other committee identified what we considered a fatal flaw, e.g., an error in a key result).

This copy’s committee reached the following decision: **Reject**

The other committee assigned to the paper recommended **Accept (Poster)**.  You can find the other set of reviews, along with any follow up discussion with the authors here:
https://openreview.net/forum?id=p9dySshcS0q